# *ANLN* Promotes the Proliferation and Migration of Gallbladder Cancer Cells via STRA6-Mediated Activation of PI3K/AKT Signaling

**DOI:** 10.3390/cancers16040752

**Published:** 2024-02-11

**Authors:** Xiang Zhu, Yong Zhang, Rui Bian, Jiyue Zhu, Weibin Shi, Yuanyuan Ye

**Affiliations:** 1Department of General Surgery, Xinhua Hospital Affiliated to Shanghai Jiao Tong University School of Medicine, No. 1665 Kongjiang Road, Shanghai 200092, China; zhusurg@163.com (X.Z.); zydl1969@sina.com (Y.Z.);; 2Shanghai Key Laboratory of Biliary Tract Disease Research, No. 1665 Kongjiang Road, Shanghai 200092, China; 3Clinical Research and Innovation Center, Xinhua Hospital Affiliated to Shanghai Jiao Tong University School of Medicine, No. 1665 Kongjiang Road, Shanghai 200092, China

**Keywords:** *ANLN*, gallbladder cancer, STRA6, PI3K/AKT

## Abstract

**Simple Summary:**

This study is the first to reveal the role and mechanism of *ANLN* (anillin, an actin-binding protein) in gallbladder cancer (GBC). Our research has led to the following conclusions: (1) *ANLN* promotes the proliferation and migration of gallbladder cancer cells. (2) Knockdown of *ANLN* promotes apoptosis and cell cycle arrest in GBC cells. (3) Knockdown of *ANLN* inhibits the Phosphatidylinositide 3-kinases (PI3K)/Serine/Threonine Kinase (AKT) signaling pathway, leading to the inhibition of GBC cell growth and migration. (4) *ANLN* activates the PI3K/AKT signaling pathway by regulating STRA6 in GBC cells.

**Abstract:**

The *ANLN* gene encodes anillin, a protein that binds to actin. Recent research has identified *ANLN*’s function in the initiation and advancement of different cancers. However, its impact on gallbladder cancer (GBC) remains unexplored. This study aimed to elucidate its possible molecular mechanisms in GBC. *ANLN* expression was assessed using quantitative real-time polymerase chain reaction (QRT-PCR), Western blotting (WB), and immunohistochemistry (IHC), revealing elevated levels in GBC tissues. *ANLN* knockdown resulted in the inhibition of cell proliferation and migration, leading to apoptosis and cell cycle arrest. Conversely, *ANLN* overexpression had the opposite effects on GBC cells. In vivo experiments confirmed that *ANLN* knockdown inhibited GBC cell growth. RNA-seq and bioinformatics analysis revealed *ANLN*’s function in activating the PI3K/AKT signaling pathway. We further confirmed that *ANLN* could upregulate STRA6 expression, which activated PI3K/AKT signaling to enhance the growth and movement of GBC cells. These findings demonstrate *ANLN*’s involvement in GBC initiation and progression, suggesting its potential as a novel target for GBC.

## 1. Introduction

Gallbladder cancer (GBC), originating from gallbladder mucosa epithelial cells, is the predominant type of biliary system cancer, making up over 70% of biliary malignancies. It is also the sixth most frequent digestive system cancer worldwide [1,2]. The most recent worldwide cancer data show that GBC had 115,949 new diagnoses and 84,695 deaths in 2020 [3]. Gallbladder tumors mostly arise in patients who have had stones for a long time or who have particular conditions such as porcelain gallbladder [4]. GBC exhibits a high degree of malignancy, with a five-year survival rate below 5% and an average survival time of only six months [5]. Surgical resection is currently the sole effective radical therapy for GBC [6,7]. However, due to its occult nature, most patients are diagnosed at an advanced stage, missing the opportunity for surgical cure. Moreover, lymph node metastasis is present in half of the patients, and those in late stages are insensitive to radiotherapy and chemotherapy [8,9]. Therefore, it is imperative to elucidate the molecular mechanisms underlying GBC onset and progression, as well as to identify novel therapeutic targets for this disease. 

The *ANLN* gene encodes the protein anillin, which was first discovered by Kathryn G. Miller et al. from Drosophila embryos in 1989. Anillin is an actin-binding protein (ABP) [10,11]. Anillin’s localization in the cell changes dynamically with the different phases of the cell cycle, indicating that anillin is essential for cell cycle regulation [12]. The *ANLN* gene could regulate cytokinesis, cell mitosis, and proliferation. Recent studies have implicated *ANLN* in various cancers, such as pancreatic cancer [13,14,15,16], gastric cancer [17], breast cancer [18,19], lung cancer [20], liver cancer [21,22,23], bladder cancer, colorectal cancer and other malignant tumors [17,24]. Research has demonstrated a gradual increase in *ANLN* expression levels from normal tissue to benign lesions and tumor tissue, indicating a close association with tumor development. Conversely, in human central nervous system tumors, *ANLN* expression is lower in tumor tissues [25]. An apparent linear correlation was found between *ANLN* expression and Ki67 (a marker of cell proliferation, involved in the regulation of chromosome segregation and regulation of mitotic nuclear division), suggesting its influence on cell proliferation. *ANLN* has been studied as a potential prognostic marker for tumors.

This study highlights *ANLN*’s crucial function in promoting the growth and metastasis of GBC. We conducted various experiments in vivo and in vitro, which indicated that downregulating *ANLN* expression levels could decrease tumor cell proliferation and migration, while inducing apoptosis and cell cycle arrest. RNA-seq and bioinformatics analysis identified STRA6 as a downstream gene of *ANLN*, and further experimentation confirmed the ability of *ANLN* to upregulate STRA6 expression, activating the Phosphatidylinositide 3-kinases (PI3K)/Serine/Threonine Kinase (AKT) signaling pathway and promoting GBC cell proliferation and migration. These results demonstrate the significant impact of *ANLN* on GBC pathogenesis and suggest that *ANLN* could be a promising diagnostic and treatment target.

## 2. Materials and Methods

### 2.1. Patient Tissue Samples

The Department of General Surgery at Xinhua Hospital, affiliated to Shanghai Jiao Tong University School of Medicine, provided us with GBC tissues and their adjacent non-tumor tissues. None of these patients had undergone any radiotherapy, chemotherapy, or immunotherapy prior to surgery. All patients gave their informed consent, and a pathologist verified all of the tissues. We collected tissue sections from a total of 29 patients. The experimental groups were divided as follows: 21 sections in the tumor group and 28 sections in the adjacent non-cancerous control group. The Research Ethics Committee at Xinhua Hospital, affiliated to Shanghai Jiao Tong University School of Medicine, authorized the use of the tissue specimens used in the experiment (ID no: XHEC-D-2023-191).

### 2.2. Immunohistochemistry (IHC)

Immunohistochemistry on GBC tissues was carried out by Wuhan Sevier Biotechnology Co., Ltd. (Wuhan, China), following the standard protocol. We fixed the GBC tissue blocks in 4% Paraformaldehyde (PFA) at 4 °C for one night. Subsequently, the samples were embedded and sectioned. The paraffin-embedded sections were first deparaffinized in xylene and subsequently rehydrated through a series of ethanol solutions with decreasing concentrations. Antigen retrieval was carried out in citrate buffer solution. To inhibit endogenous peroxidase activity, sections were incubated in 3% hydrogen peroxide solution in the dark for 25 min. This step was followed by a 30-min block at room temperature using 3% BSA. The slides were then incubated with primary antibodies against ki67 (ab16667, Abcam, Cambridge, UK, 1:100) and *ANLN* (ab211872, Abcam, UK, 1:100) at 4 °C overnight. The following day, they were incubated with enzyme-labeled secondary antibodies. Finally, the slides were developed with diaminobenzidine (DAB) and counterstained with hematoxylin. After dehydration and mounting, images were captured using a Leica scanner. Hematoxylin stained the nuclei blue, while positive expression using DAB appeared as brownish-yellow.

### 2.3. Cell Culture and Transfection

In this experiment, we used GBC-SD, SGC996, EHGB1, and NOZ cells. All cell lines were obtained from Shanghai Key Laboratory of Biliary Tract Disease Research. All the cells were cultured in Dulbecco’s Modified Eagle Medium (DMEM, Gibco, NY, USA) supplemented with 10% fetal bovine serum and 1% penicillin streptomycin (Yeasen, Shanghai, China) and placed in a cell incubator with 5% carbon dioxide (CO_2_) at 37 °C. The cells were passaged when the confluence exceeded 70%.

We knocked down *ANLN* in GBC-SD and SGC996 cells using RFect transfection reagent (Baidai Biotech, Changzhou, China). The cell transfection experiment was conducted according to the manufacturer’s instructions. Small interfering RNAs (siRNAs) were synthesized by Beijing Tsingke Biotech. The siRNAs are detailed in Table 1.

We followed the instructions to carry out the transfection procedure and verified the knockdown efficiency using qPCR and WB. We overexpressed *ANLN* in NOZ cells using the human mRNA sequence synthesized by Beijing Tsingke Biotech base on the NCBI website (https://www.ncbi.nlm.nih.gov/, accessed on 1 February 2024) and constructed it into the PLVX lentiviral vector. The PLVX vector without any contents was utilized as the control group for comparison. The overexpression efficiency was verified using qPCR and WB after lentiviral transduction.

### 2.4. Lentivirus Packaging and Infection

For lentivirus packaging, we used Yeasen liposome transfection reagent along with pMD2.G and psPAX2 (Addgene, Watertown, MA, USA). These were co-transfected into 293FT cells along with the target plasmid at a ratio of 1.25:3.75:5. Plasmid transfection experiments were performed according to instructions. We collected the cell supernatant at 48 h and 72 h, then filtered and concentrated it. The precipitate was collected after centrifugation (3750× *g*, 60 min, 4 °C) the next day. During cellular infections, in the culture dish, polybrene (Yeasen, Shanghai, China) was added at the dosage recommended in the instruction manual. After 24 h, the serum-free culture medium was replaced with complete culture medium.

Following this, we performed two weeks of selection with puromycin. We confirmed the overexpression efficiency using qPCR and WB.

### 2.5. Total RNA Extraction and qRT-PCR

When the cells reached more than 70% confluence in the culture dish, we extracted RNA using an EZ-press RNA Purification Kit, following the manufacturer’s instructions. After extraction, we immediately placed the RNA on ice and measured the concentration. The reverse transcription reagent was acquired from TaKaRa, following the manufacturer’s instructions. We used SYBR Green (Yeasen) to perform qPCR on a QuantStudio 3 real-time thermocycler device (Applied Biosystems, Waltham, MA, USA). We used Glyceraldehyde-3-phosphate dehydrogenase (GAPDH) as the internal reference gene for this study, and the primer information is presented in Table 2 below.

### 2.6. Western Blot

For cell protein extraction, the lysate was SDS lysate with added protease inhibitors and phosphatase inhibitors (EpiZyme, Shanghai, China). Proteins were quantified using a BCA kit (EpiZyme). In each lane, 20 µg of protein lysate was loaded; for phosphorylated proteins with a lower abundance, 40 µg of protein lysate per lane was introduced. We used SDS-PAGE (EpiZyme) gels to separate the prepared protein samples and subsequently transferred them onto 0.25 μm polyvinylidene fluoride (PVDF) membranes (Merck Millipore, Darmstadt, Germany) using electroblotting. After that, we blocked the membranes in 5% milk for one hour, rinsed them three times with Tris Buffered Saline with Tween (TBST) (Servicebro, Wuhan, China), and exposed them to the corresponding primary antibody overnight. The following day, we washed the membranes three times with TBST and then treated them with the Horseradish peroxidase (HRP)-conjugated secondary antibodies (1:5000, #SA00001-2; #SA00001-1; proteintech) for one hour. The bands were then detected using chemiluminescent detection reagents (Beyotime, Shanghai, China). Information on the primary antibodies is shown in Table 3.

### 2.7. CCK8 Assay and Colony Formation Assay

In this experiment, the changes in the cell proliferation level were measured using CCK8 assay. The reagent used was CCK8 reagent from Yeasen. In a 96-well plate, 2000 GBC cells were seeded per well and incubated until they adhered to the plate. Then, we followed the instructions to add CCK8 reagent. The concentration of the CCK8 reagent was 10%. After the addition of the CCK8 reagent, the samples were incubated for 2 h in a cell culture incubator at 37 °C with 5% CO_2_. We measured the absorbance at 450 nm.

For the colony formation assay, a 6-well plate was used with 1000 GBC cells seeded in each well and cultured for two weeks. Then, we used paraformaldehyde to fix them and crystal violet to stain them. Upon completion of the colony formation assay, photographs were taken, followed by the quantification of colony numbers using ImageJ software (1.54d). 

### 2.8. Transwell Migration Assay and Wound Scratch Assay

In this experiment, the changes in GBC cell migration were measured using transwell assay. The transwell chamber was acquired from Corning, with a pore size of 8 um. The cells were digested, resuspended, counted and plated. The lower chamber was filled with a medium supplemented with ten percent serum, while cells were seeded in the upper chamber at a density of 2 × 10^4^ cells in serum-free medium. After being cultured for a period of 24 h, the specimens were subsequently fixed and stained. The resulting images were captured of the center and the four corners of the transwell chamber. The imaging equipment used was a LEICA microscope. 

The changes in GBC cell migration ability were assessed using scratch assay. The cells were plated at a 30% density in a 6-well plate and cultured. Next, they were rinsed with PBS and then changed to DMEM medium without serum. A scratch was made and photographed. After 24 h, the photographs were taken again and the changes in scratch width were observed. The wound healing assay images were taken from a fixed position, marked by horizontal lines drawn on the back of the six-well plate. The measurement of the wound healing distance and the counting in the transwell assay were both performed using ImageJ software.

### 2.9. Subcutaneous Xenograft Models

BALB/C nude mice, aged 4–6 weeks old, were used in this experiment. They were bred in a specific pathogen free (SPF) environment and allowed to acclimate for one week. Six nude mice were assigned to each group, using two GBC-SD cell lines with knockdown of *ANLN* and a negative control group that were successfully constructed before. The GBC-SD cells were inoculated into the axillary region of the nude mice at a concentration of 2 × 10^5^ cells per mouse. Tumor formation was observed after one week. We euthanized the mice two weeks later and removed and weighed the subcutaneous tumors. The following formula was used to calculate the tumor volume: (V = 1/2ab^2^). We fixed the tumor tissues with paraformaldehyde and then embedded, sectioned and stained them with hematoxylin eosin staining (HE) and immunohistochemistry. 

### 2.10. mRNA Sequencing and Bioinformatic Analysis

Total RNA in GBC-SD cell negative control and *ANLN* knockdown groups was extracted using the TRIzol reagent (Yeasen). LC Bio Technology Co., Ltd. (Hangzhou, China) provided the sequencing service and subsequent analysis using the Illumina NovaseqTM 6000 platform.

### 2.11. Flow Cytometry: Apoptosis Assay and Cell Cycle Assay

In the cell apoptosis experiment, a BD Cell Apoptosis Kit was utilized. After treating the cells in a six-well plate, three rounds of PBS washing were performed, followed by digestion using trypsin without Ethylenediaminetetraacetic Acid (EDTA). The cells were then centrifuged, resuspended, and sequentially stained with Propidium Iodide (PI) and 7 AminoactinomycinD (AAD) from the cell apoptosis kit. After ten minutes of incubation, the cells were analyzed using flow cytometry.

A Cell Cycle Kit (Beyotime) was employed in this experiment. One million cells were collected, resuspended in precooled 70% ethanol, and stored at 4 °C overnight. The following day, the cells were centrifuged, resuspended, stained with PI, and incubated for 30 min at 37 °C. Flow cytometry was utilized to determine the cell cycle distribution.

### 2.12. Statistical Analysis

The *p*-values were calculated using GraphPad Prism software (8.0.1). For comparing the distinction between two groups, an unpaired two-sided Student t-test was employed. Alternatively, for comparing distinctions among more than two groups, one-way analysis of variance (ANOVA) followed by Tukey’s multiple comparisons or two-way ANOVA followed by Tukey’s multiple comparisons were used. (*p* < 0.05 *, *p* < 0.01 **, *p* < 0.001 ***, *p* < 0.0001 ****).

## 3. Results

### 3.1. Expression of ANLN Is Upregulated in Human GBC Tissues and GBC Cell Lines 

Microarray analysis demonstrated a significant increase in *ANLN* mRNA expression levels in GBC tissues when compared to adjacent non-tumor tissues. To validate these findings, qRT-PCR and Western blotting were employed to assess *ANLN* levels in GBC cell lines. The results demonstrated higher *ANLN* mRNA expression in GBC-SD and SGC996 cells, and lower expression in NOZ and EHGB1 cells, consistent with the protein levels (Figure 1A,B). Additionally, immunohistochemistry performed on matched GBC and adjacent normal tissues showed higher *ANLN* levels in tumor tissues (Figure 1C). These findings collectively indicate elevated expression of *ANLN* in both human GBC cell lines and tissues. 

### 3.2. Knockdown of ANLN Attenuates GBC Cell Proliferation and Migration

Based on the *ANLN* expression in GBC cell lines, we chose to perform knockdown experiments on GBC-SD and SGC996 cells, which exhibited relatively high *ANLN* expression. We designed four small interfering RNAs (siRNAs) targeting *ANLN* and verified their knockdown effects in GBC-SD and SGC996 cells. The findings revealed a significant reduction in *ANLN* expression at both the mRNA and protein levels following siRNA transfection. Subsequently, two siRNAs exhibiting superior efficacy were selected for further functional experiments (Figure 2A,B). The CCK8 assay revealed a reduction in the growth of GBC-SD and SGC996 cells following *ANLN* knockdown (Figure 2C). Additionally, the colony formation assay demonstrated a marked inhibition in cell colony formation after *ANLN* silencing (Figure 2D). These observations collectively suggest a role for the *ANLN* gene in promoting cell proliferation in GBC cell lines. Subsequently, the wound healing assay showed a decreased wound healing speed in GBC-SD and SGC996 cells following *ANLN* knockdown, compared to the control group (Figure 2E). Furthermore, the transwell assay results confirmed a notable reduction in the movement ability of GBC cells after *ANLN* silencing (Figure 2F). These findings collectively suggest that the *ANLN* gene enhances the migration of GBC cells in vitro. 

### 3.3. Knockdown of ANLN Promotes Apoptosis and Cell Cycle Arrest in GBC Cells

In order to further elucidate the potential pathogenic mechanism of *ANLN*, flow cytometry was utilized to assess changes in apoptosis and cell cycle in GBC cells flowing *ANLN* knockdown. The cell cycle assay, employing PI staining, revealed a significant increase in the number and proportion of GBC-SD and SGC996 cells in the G2 phase after *ANLN* silencing, indicating cell cycle arrest in the G2/M phase (Figure 3A). Additionally, *ANLN* knockdown was found to promote cell apoptosis in GBC cells compared to the negative control group (Figure 3B). These findings suggest that *ANLN* knockdown induces cell cycle arrest and apoptosis in GBC cells.

**Figure 1 cancers-16-00752-f001:**
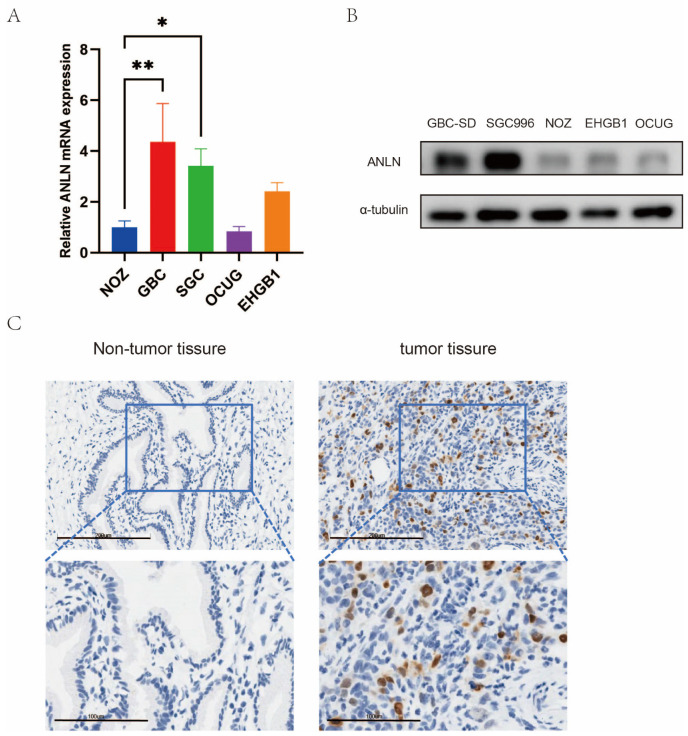
Expression of *ANLN* in human gallbladder cancer (GBC). (**A**,**B**) *ANLN* mRNA and protein expression in GBC cell lines (NOZ, GBC-SD, SGC996, OCUG, EHGB1). (**C**) Representative immunohistochemistry (IHC) images of GBC tissues and their corresponding adjacent tissues (scale bar: 200 μm; amplified image: 100 μm). The uncropped blots are shown in Appendix A. *p* < 0.05 *, *p* < 0.01 **.

**Figure 2 cancers-16-00752-f002:**
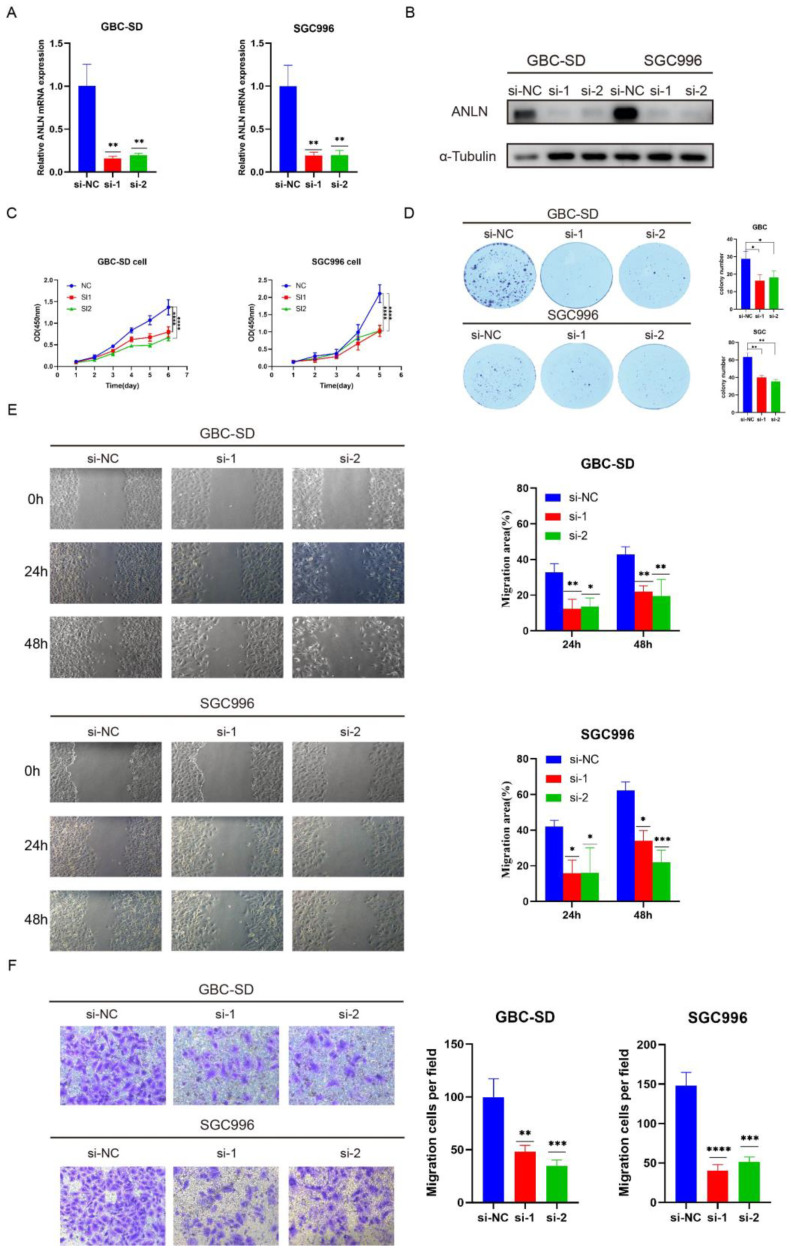
Knockdown of *ANLN* attenuates GBC cell proliferation and migration. (**A**,**B**) *ANLN* mRNA and protein expression levels were assessed in GBC-SD and SGC996 cells following si-*ANLN* transfection. (**C**) The CCK-8 assay was utilized to quantify the growth of GBC cells (GBC-SD, SGC996) following si-*ANLN* transfection. (**D**) The colony formation assay was performed to evaluate the impact of *ANLN* knockdown on GBC cell (GBC-SD, SGC996) colony formation, with statistical significance determined via colony count analysis. (**E**) The migration abilities of GBC cells (GBC-SD, SGC996) were assessed using the wound healing assay, and statistical significance was determined by analyzing the area of cell migration. Magnification: 200×. (**F**) The transwell assay was employed to measure the migration of treated GBC-SD and SGC996 cells, with statistical significance analyzed based on the number of cells invaded. Magnification: 200×. The uncropped blots are shown in Appendix A. *p* < 0.05 *, *p* < 0.01 **, *p* < 0.001 ***, *p* < 0.0001 ****.

**Figure 3 cancers-16-00752-f003:**
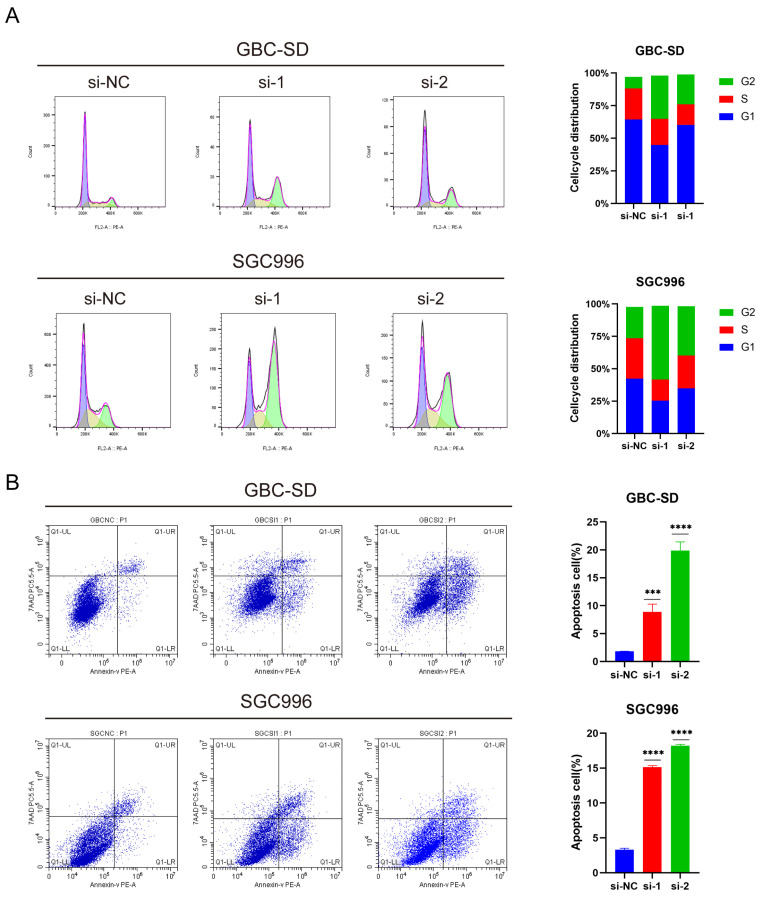
*ANLN* knockdown leads to apoptosis and cell cycle arrest in GBC cells (GBC-SD, SGC996). (**A**) Cell cycle distribution with *ANLN* silencing. Purple: G1 phase; Yellow: S phase; Green: G2 phase.(**B**) Flow cytometry assessment of apoptosis rates through annexin V-Phycoerythrin (PE)/7-AminoactinomycinD (AAD) staining. *p* < 0.001 ***, *p* < 0.0001 ****.

### 3.4. ANLN Facilitates GBC Tumor Growth In Vivo

Building upon the findings from the cell experiments, we conducted further investigations into the role of *ANLN* in an in vivo setting. We designed shRNA targeting *ANLN* and verified its knockdown effect in the GBC-SD cell line. Subsequently, stable GBC-SD cells with *ANLN* silencing and negative control cells were subcutaneously implanted into the armpits of nude mice to establish a subcutaneous tumor transplantation model. The results are shown in Figure 4A, revealing significant inhibition of GBC tumor growth in the nude mouse subcutaneous tumor model following *ANLN* silencing, as confirmed by the reduced weight and volume of the subcutaneous tumors (Figure 4B,C). Additionally, immunohistochemical analysis demonstrated a decrease in Ki67 levels in the subcutaneous tumor tissues after *ANLN* silencing compared to the negative control group (Figure 4D). These findings collectively support the role of *ANLN* in promoting GBC growth in vivo.

### 3.5. Knockdown of ANLN Inhibited the PI3K/AKT Signaling Pathway, Leading to Inhibition of GBC Cell Growth and Migration

To delve deeper into the specific molecular mechanism of *ANLN* in GBC, we performed mRNA sequencing on GBC-SD cells with *ANLN* knockdown or a negative control. Bioinformatic analysis was performed based on statistical criteria of |log2FC| ≥ 1 and Q-value ≤ 0.05. Volcano plots (Figure 5A) revealed 864 upregulated genes and 1147 downregulated genes after *ANLN* silencing compared to the negative control group. The differentially expressed genes (DEGs) were visualized using a heatmap (Figure 5B). Additionally, Gene Ontology (GO) analysis demonstrated enrichment of the DEGs in protein binding as well as the cytoplasm, cytosol, and cytoskeleton (Figure 5C). Furthermore, Kyoto Encyclopedia of Genes and Genomes (KEGG) analysis indicated enrichment of the DEGs in pathways related to cancer, small cell lung cancer, and the PI3K/AKT signaling pathway (Figure 5D). These KEGG analysis results suggested a potential link between the PI3K/AKT signaling pathway and the impact of *ANLN* on GBC cell proliferation and migration. 

The PI3K/AKT signaling pathway, a well-established oncogenic pathway, plays a crucial role in cell metabolism, proliferation, growth, and regulatory processes associated with cancer initiation, chemotherapy resistance, and angiogenesis. Our study revealed that *ANLN* expression influences the levels of P-PI3K and P-AKT in GBC-SD and SGC996 cells. Figure 5E demonstrates that knockdown of *ANLN* led to decreased expression of P-PI3K and P-AKT, while the levels of PI3K and AKT expression remained unchanged. These results suggest that *ANLN* contributes to the activation of the PI3K/AKT signaling pathway in GBC cells. To further elucidate these findings, we utilized the LY294002 PI3K inhibitor and conducted a rescue experiment in NOZ cells with *ANLN* overexpression. The results of the WB experiment are depicted in Figure 5F, revealing that LY294002 inhibited the cell proliferation and migration induced by *ANLN* overexpression (Figure 6A–C). This suggests that *ANLN* regulates the proliferation and migration of GBC cells through the PI3K/AKT signaling pathway.

### 3.6. ANLN Activates the PI3K/AKT Signaling Pathway by Regulating STRA6

Upon analyzing the RNA sequencing results, GO analysis unveiled enrichment of genes with distinct expression levels primarily associated with protein interaction. Notably, our attention was drawn to the significantly downregulated gene STRA6, involved in protein binding and responsible for facilitating retinol separation and its transportation across the cell membrane. We detected the expression of STRA6 using qPCR and found that it was decreased by *ANLN* knockdown (Figure 6D). Consistent results were observed at the protein level, further confirming *ANLN*’s regulatory influence on STRA6 expression (Figure 6E). In GBC-SD cells, silencing STRA6 led to reduced levels of P-PI3K and P-AKT, while the expression of PI3K and AKT remained unaltered (Figure 6F), indicating that STRA6 depletion inhibited the PI3K/AKT signaling pathway.

**Figure 6 cancers-16-00752-f006:**
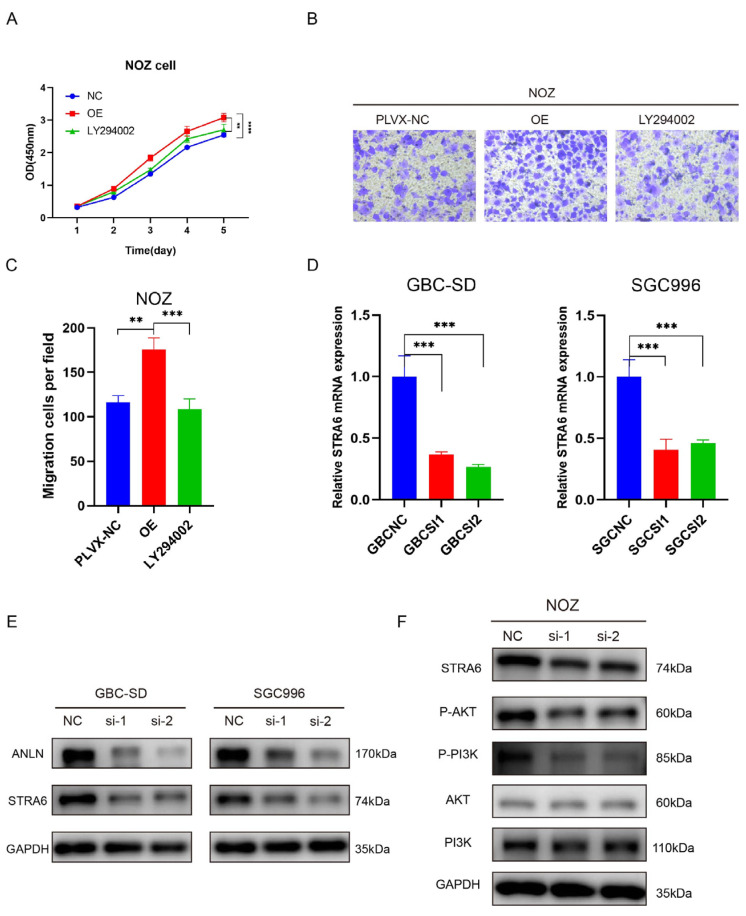
*ANLN*-mediated activation of the PI3K/AKT signaling pathway through STRA6 regulation. (**A**) The proliferation ability of *ANLN* overexpression GBC cells (NOZ) without or with the PI3K inhibitor LY294002 (20 μM, 24 h) was determined using CCK-8 assays. (**B**,**C**) The transwell assay was conducted to evaluate the migratory ability of NOZ cells. The experimental groups were as follows: PLVX-NV as the control group, the *ANLN* overexpression group, and the group with *ANLN* overexpression followed by the addition of LY294002 to block the PI3K/AKT signaling pathway. (**D**,**E**) Analysis of STRA6 expression in *ANLN*-silenced GBC cells (GBC-SD, SGC996) using qRT-PCR and WB. (**F**) WB results depicting the expression levels of p-PI3K, PI3K, p-AKT, and AKT in STRA6-silenced NOZ cells. The uncropped blots are shown in Appendix A. *p* < 0.01 **, *p* < 0.001 ***, *p* < 0.0001 ****.

## 4. Discussion

Gallbladder cancer is a highly aggressive and poor-prognosis malignant tumor. The incidence and mortality of GBC have been increasing in recent years [26,27]. Due to the limited treatment options and poor therapeutic effects [28,29], there is a critical need to elucidate the molecular mechanisms underlying GBC development. Therefore, the objective of this study was to examine the role of the *ANLN* gene in GBC. *ANLN* encodes anillin involved in cell growth, migration, and cytokinesis [30,31]. Recent studies have revealed the involvement of *ANLN* in the initiation and progression of various cancer types. For example, in lung cancer, the knockdown of *ANLN* can activate pyroptosis and inhibit the progression of lung adenocarcinoma [32]. Similarly, elevated *ANLN* expression in liver tumors is linked to poor prognosis, and *ANLN* knockdown has been demonstrated to inhibit the occurrence and progression of liver tumors by influencing cytokinesis impairment. *ANLN* with the m6A modification has a crucial function in liver cancer bone metastasis [21,33,34,35]. Cao et al. reported that inhibiting the USP10-*ANLN* axis hinders the cell cycle progression of esophageal squamous cell carcinoma (ESCC) [36]. Our study revealed elevated *ANLN* expression levels in GBC tissues. Upon silencing *ANLN* in GBC-SD and SGC996, we observed inhibited cell proliferation, migration and cell division, as well as increased cell apoptosis. These results indicate that *ANLN* functions as a novel oncogene in human GBC.

The PI3K/AKT pathway is a well-known signaling pathway that has a significant impact on tumor biology processes, regulating various vital physiological events including tumor growth, invasion and metastasis [37,38,39,40,41]. Abnormal activation of the PI3K/AKT pathway is closely linked to the onset and advancement of GBC. Existing studies have reported that in gallbladder cancer tissues and cells, upstream or downstream molecules of PI3K/AKT signaling pathway, such as EGFR, HER2, and PTEN, etc. [42,43,44], experience gene mutations, amplifications or expression dysregulation. These changes lead to sustained activation of the PI3K/AKT signaling pathway, consequently promoting proliferation, anti-apoptotic processes, invasion, metastasis, angiogenesis, and drug resistance in GBC cells. Therefore, the development of inhibitors targeting the PI3K/AKT signaling pathway is still under investigation as a novel therapeutic approach for GBC [41,45,46,47]. AKT can modulate the ataxia telangiectasia-mutated gene (ATM)/ATM and Rad3-related (ATR) pathways, which are critical for the cellular response to DNA damage [48]. Through phosphorylation and interaction with these kinases, AKT can influence the repair of DNA double-strand breaks and replication stress [49]. Chen et al. discovered that knocking down *ANLN* in liver cancer cells results in the occurrence of DNA damage [21].The upregulation of AKT often leads to enhanced cell survival and proliferation, even in the presence of DNA damage, contributing to genomic instability [50]. This instability is a hallmark of cancer cells and is associated with increased mutation rates and chromosomal abnormalities. However, chronic overactivation of AKT can result in inadequate or erroneous DNA repair, further exacerbating genomic instability. In the context of AKT upregulation, p53′s function can be compromised. AKT can negatively regulate p53 through direct phosphorylation or through downstream effectors, leading to decreased apoptosis and increased cell survival. The dysregulation of this balance due to AKT overexpression can lead to unchecked cell proliferation and survival, contributing to tumorigenesis. The interplay between AKT and p53 is complex and significant in cancer biology [51,52], as alterations in their regulation can lead to resistance to chemotherapy and radiotherapy. In this study, we performed KEGG analysis on the transcriptome sequencing data of GBC-SD cells with *ANLN* knockdown and discovered that the PI3K/AKT signaling pathway was suppressed by *ANLN* knockdown. To further validate the impact on the PI3K/AKT signaling pathway, we evaluated the protein expression levels of PI3K, P-PI3K, AKT, and P-AKT in *ANLN*-depleted GBC cells, observing a reduction in P-PI3K and P-AKT levels. Conversely, the levels of PI3K and AKT proteins remained relatively constant, indicating that *ANLN* knockdown inhibited the PI3K/AKT signaling pathway. After overexpressing *ANLN*, the results were the opposite to the above. We then added the inhibitor LY2900000 of PI3K/AKT to the cells overexpressing *ANLN*, and we observed inhibition of the elevated P-PI3K and P-AKT protein levels. The findings from our experiments indicate that *ANLN* functions as an oncogene in GBC through the PI3K/AKT signaling pathway. Consequently, blocking the PI3K/AKT signaling pathway can potentially counteract the cancer-promoting effects of *ANLN* in GBC cells.

STRA6, a membrane protein, plays a vital role in retinol metabolism and facilitates the transportation of retinol into cells [53]. Recent research has uncovered the involvement of STRA6 in tumor initiation and progression as well as that STRA6 is highly expressed in various tumor cells, such as thyroid cancer [54,55], colorectal cancer [56], lung cancer [57], gastric cancer [20,58], etc., and is associated with prognosis. He et al. found that STRA6 can promote the progression of thyroid cancer through the ILK/AKT/mTOR axis [55]. In this study, mRNA sequencing of GBC-SD cells following *ANLN* knockdown revealed significant downregulation of STRA6. *ANLN* knockdown resulted in a reduction in STRA6 levels, which was confirmed using both qPCR and Western blotting. To investigate the impact of STRA6 on the PI3K/AKT signaling pathway, we measured the protein levels of PI3K, P-PI3K, AKT, and P-AKT using Western blotting. We observed a decrease in P-PI3K and P-AKT protein levels following STRA6 knockdown, while the levels of PI3K and AKT remained unchanged. These results suggest that STRA6 knockdown can inhibit the PI3K/AKT signaling pathway. Therefore, *ANLN* may regulate STRA6 to activate the PI3K/AKT signaling pathway, ultimately promoting the proliferation and migration of GBC cells.

## 5. Conclusions

In conclusion, our study illustrated elevated *ANLN* expression in GBC tissues and its role in promoting cancer through the activation of the PI3K/AKT signaling pathway. We also elucidated the regulatory role of *ANLN* in STRA6 expression, leading to the activation of the downstream PI3K/AKT signaling pathway and the enhancement of GBC cell proliferation and migration. Consequently, our findings underline that *ANLN* could be a promising therapeutic target for GBC.

## Figures and Tables

**Figure 4 cancers-16-00752-f004:**
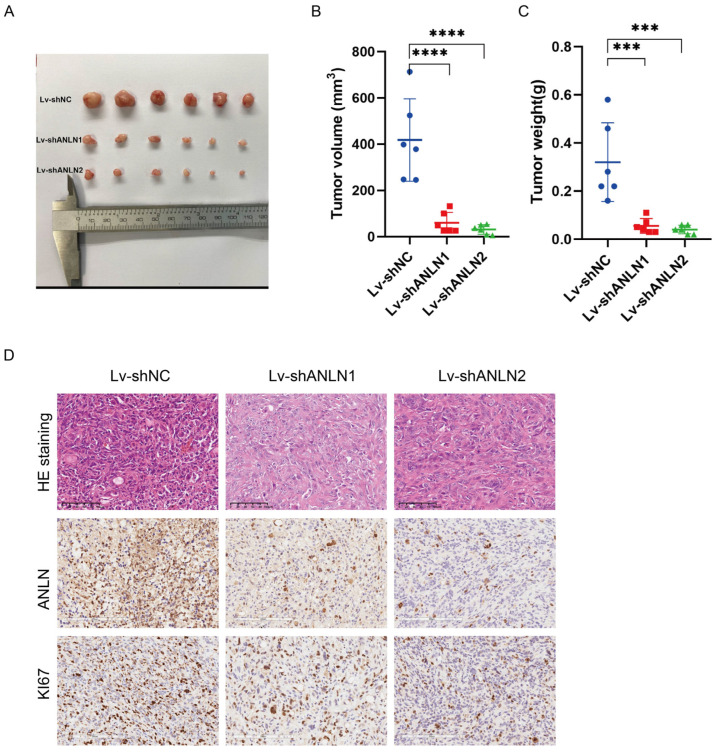
*ANLN* facilitates GBC tumor growth in vivo. (**A**) Tumor growth is inhibited in vivo following *ANLN* knockdown. (**B**,**C**) Tumor volumes and weights in the negative control group and *ANLN*-knockdown groups. (**D**) Representative images of hematoxylin eosin (HE) staining and immunohistochemistry for *ANLN* and Ki67. *p* < 0.001 ***, *p* < 0.0001 ****.

**Figure 5 cancers-16-00752-f005:**
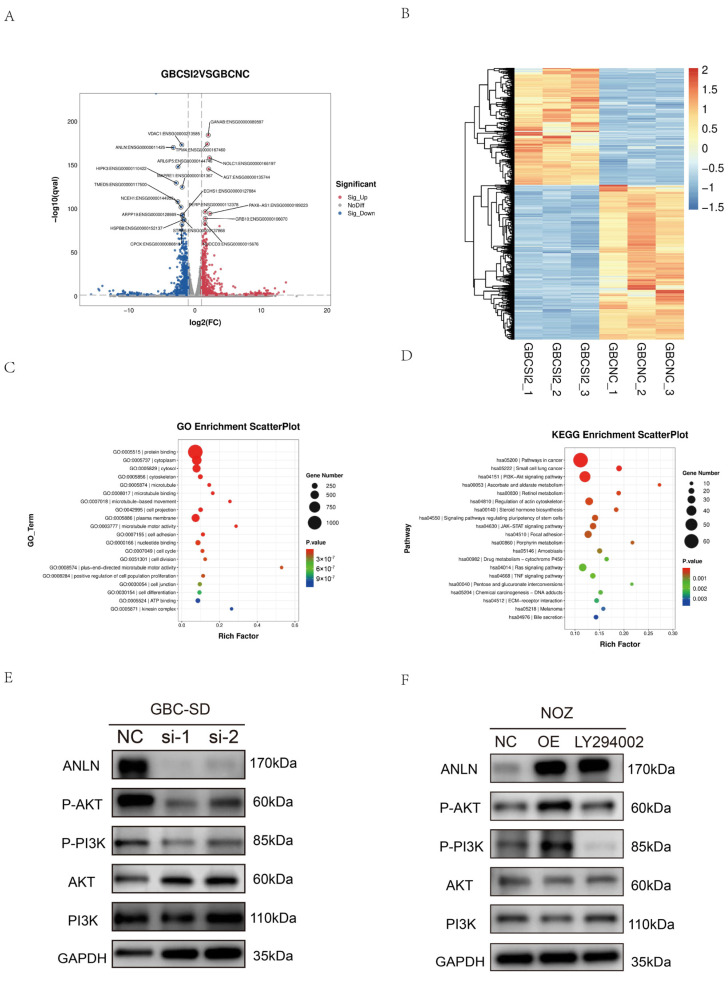
RNA-seq result analysis and protein level detection. (**A**) Volcano plots of differentially expressed genes (DEGs) after silencing *ANLN* in GBC-SD cells. (**B**). Heat map of DEGs. (**C**,**D**). Gene Ontology (GO) and Kyoto Encyclopedia of Genes and Genomes (KEGG) analysis of DEGs. (**E**,**F**) WB results of *ANLN*, Phosphorylation Phosphotylinosital 3 kinase (p-PI3K), Phosphatidylinositol 3-kinase (PI3K), phosphorylated AKT protein (p-AKT) and AKT (Serine/Threonine Kinase) expression in GBC cells (LY294002 is a broad-spectrum PI3K inhibitor with PI3K as its target). The uncropped blots are shown in Appendix A.

**Table 1 cancers-16-00752-t001:** Small interfering RNA (siRNA) information. (ANLN: Anillin, actin binding protein; STRA6: Stimulated by Retinoic Acid 6).

	Sense 5′-3′	Antisense 5′-3′
*ANLN*-si1	GCAUCUGCUAGGAUCAAUA	UAUUGAUCCUAGCAGAUGC
*ANLN*-si2	CAGUGAUGUCCUAGAGGAA	UUCCUCUAGGACAUCACUG
STRA6-si1	GCUCUGGAAGUGUGCUACA	UGUAGCACACUUCCAGAGC
STRA6-si2	CCAAGAUCUACAAGUACUA	UAGUACUUGUAGAUCUUGG

**Table 2 cancers-16-00752-t002:** Primer information (GAPDH: Glyceraldehyde-3-phosphate dehydrogenase).

	Forward	Reverse
*ANLN*	TGCACCATTGGCACAAACAG	CCAGATTCAGCTCGAGGGAC
STRA6	AGACCAGGTCCCACACTGA	TTCATAATAGCCAAAGGCATAAAAA
GAPDH	GGAGCGAGATCCCTCCAAAAT	GGCTGTTGTCATACTTCTCATGG

**Table 3 cancers-16-00752-t003:** Primary antibody information (PI3K: Phosphatidylinositol 3-kinase; P-PI3K: Phosphorylation Phosphotylinosital 3 kinase; AKT: Serine/Threonine Kinase; P-AKT: phosphorylated AKT protein).

Antibody	Application	Dilution	Company
*ANLN*	IHC	1:100	Abcam (Cambridge, UK)
Ki67	IHC	1:100	Cell Signaling Technology (Danvers, MA, USA)
*ANLN*	WB	1:1000	Abcam (Cambridge, UK)
*ANLN*	WB	1:1000	Proteintech (Wuhan, China)
STRA6	WB	1:1000	proteintech (Wuhan, China)
PI3K	WB	1:10,000	ABclonal (Wuhan, China)
P-PI3K	WB	1:1000	ABclonal (Wuhan, China)
Pan-AKT	WB	1:2000	ABclonal (Wuhan, China)
P-AKT	WB	1:2000	ABclonal (Wuhan, China)
GAPDH	WB	1:25,000	proteintech (Wuhan, China)
α-tubulin	WB	1:20,000	proteintech (Wuhan, China)

## Data Availability

Data are available on request due to restrictions, e.g., privacy or ethical considerations. The data presented in this study are available on request from the corresponding author.

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
