# Peer review of "ANLN* Promotes the Proliferation and Migration of Gallbladder Cancer Cells via STRA6-Mediated Activation of PI3K/AKT Signaling"

_cancers, 2024, doi:10.3390/cancers16040752_

Round 1

Reviewer 1 Report

Comments and Suggestions for Authors

Involving study on the molecular genesis of gallbladder cancer. Researchers ask the question of how important anillin may be in the genesis of gallbladder cancer; but ultimately in different types of neoplasms located in the digestive system, not having detected high anillin values in tumors of the nervous system. We agree with the initial discussion of the introduction, although we must ask to add, as a clinician, that gallbladder tumors mostly arise in patients who have had stones for a long time or who have particular forms such as porcelain gallbladder (doi.org/10.1007/s10353-021-00710-2 to be cited in the bibliography). Such patients are subjected to chronic irritative/inflammatory stimulation which stimulates the interleukin cascade which in turn can induce the production of anillin and cell growth with subsequent modifications well described by colleagues. The laboratory study conducted on human cells was excellent, first with cultivation and then with the extraction of DNA from patients and the results were convincing with the reduction of anillin production and the subsequent reduction of cell growth. The researchers completely answer the question they initially asked themselves with irrefutable tests. This study may have future clinical implications. Excellent tables and figures with excellent explanations. The bibliography is relative and supports the initial thesis. English is excellent

Author Response

Dear Reviewer,

First and foremost, I would like to express my sincere gratitude for the time and effort you have dedicated to reviewing our manuscript. Your insightful comments and suggestions are invaluable to our research and will guide us in further improving the quality of our paper.

In response to your valuable suggestion, we have added content regarding the etiology of gallbladder cancer in the first part of the introduction of our article. The following text has been incorporated and highlighted in the manuscript: "Gallbladder tumors mostly arise in patients who have had stones for a long time or who have particular forms such as porcelain gallbladder." We have included the following reference in our manuscript as per your suggestion: doi.org/10.1007/s10353-021-00710-2.

Once again, thank you for your meticulous review and constructive feedback. We look forward to your re-evaluation and hope to receive further guidance from you.

Kind regards,

Xiang Zhu

Department of General Surgery, Xinhua Hospital Affiliated to Shanghai Jiao Tong

University School of Medicine, No. 1665 Kongjiang Road, Shanghai 200092, China;

Reviewer 2 Report

Comments and Suggestions for Authors

Please refer to the attached PDF document for my comments. 

Comments on the Quality of English Language

Moderate editing of the language is required. Numerous grammatical and syntax errors were found throughout the manuscript. However, they did not interfere with my understanding of the manuscript. I would suggest a thorough revision of the manuscript by a qualified individual. 

Author Response

Dear Reviewer

I would like to extend my deepest gratitude for the thorough and insightful review you provided for our manuscript. Your expertise and detailed feedback have been instrumental in enhancing the quality and clarity of our work. We have meticulously reviewed each of your comments and suggestions, and have made substantial revisions to our manuscript accordingly.

In response to your valuable feedback, we have made revisions point by point. A detailed reply to each of your comments is provided in the attached PDF file. Corresponding modifications have been made in the manuscript, and these changes are highlighted for ease of review.

I greatly appreciate the time and effort you have invested in reviewing our work. Your constructive critique is not only invaluable for this manuscript but also beneficial for our future research endeavors. We eagerly anticipate your re-review and any further recommendations you may have.

With warm regards,

Xiang Zhu

Department of General Surgery, Xinhua Hospital Affiliated to Shanghai Jiao Tong

University School of Medicine, No. 1665 Kongjiang Road, Shanghai 200092, China;

Reviewer 3 Report

Comments and Suggestions for Authors

Manuscript entitled „ ANLN Promotes the Proliferation and Migration Of Gallbladder Cancer Cells Vis STRA6-Mediate Activation Of PI3K/AKT Signaling” is very interesting, well-written and well-planned experimental work. I fully support the publication of this manuscript; however I recommend the major revision of manuscript (especially the part Materials and Methods). Some corrections should be made to the text according to the following comments:

Simple Summary:

Line 15, 18, - explain in full name an abbreviation ANLN, GBC, PI3K/AKT

Introduction

Line 38 - explain in full name an abbreviation GBC

Line 53 – put a dot after [11]

Line 61 – explain what Ki67 means.

Materials and Methods

2.1. Patient tissue samples

How many samples and from how many patients were used to collect the results in each experimental group?

2.2. Immunohistochemistry (IHC)

Line 85 - explain in full name an abbreviation PFA

Please describe in detail how the immunohistochemical staining procedure was performed: how the material for staining was prepared, what type of antibodies were used, in what dilutions, etc.

What techniques were used to collect the results obtained using immunohistochemical staining?

2.5. Total RNA extraction and qRT-PCR

Line 123 - explain in full name an abbreviation GAPDH

Line 135 - explain in full name an abbreviation PVDF

Line 137 - explain in full name an abbreviation TBST

Line 139 - explain in full name an abbreviation HRP

Tables 1,2,3 - Acronyms/Abbreviations/Initialisms should be defined the first time they appear in each of three sections: the abstract; the main text; the first figure or table. When defined for the first time, the acronym/abbreviation/initialism should be added in parentheses after the written-out form.

Line 164 - explain in full name an abbreviation PBS

2.9. Subcutaneous xenograft models:

Line 171 - explain in full name an abbreviation SPF

Line 178 - explain in full name an abbreviation HE

2.11. Flow cytometry: Apoptosis assay, Cell cycle assay

Line 190 - explain in full name an abbreviation EDTA

Line 191 - with PI and 7AAD from the cell apoptosis kit – explain abbreviations

Author Response

Dear Reviewer

I am profoundly thankful for the time, effort, and expertise you have invested in reviewing our manuscript. Your detailed and constructive comments have been invaluable in enhancing the overall quality and rigor of our research.

In response to your valuable feedback, we have made revisions point by point. A detailed reply to each of your comments is provided in the attached PDF file. Corresponding modifications have been made in the manuscript, and these changes are highlighted for ease of review.

Thank you once again for your constructive feedback and for considering our manuscript for publication. We eagerly anticipate your re-review and any further recommendations you may have.

Warm regards,

Xiang Zhu

Department of General Surgery, Xinhua Hospital Affiliated to Shanghai Jiao Tong

University School of Medicine, No. 1665 Kongjiang Road, Shanghai 200092, China;

Round 2

Reviewer 3 Report

Comments and Suggestions for Authors

Dear Author, thank you very much for responding to my comments in the revised version of the manuscript. In its current form, the manuscript is suitable for publication.